# Recruiting and retaining bachelor qualified nurses in German hospitals (BSN4Hospital): protocol of a mixed-methods design

Claudia B Maier ![ORCID],[1,2] Julia Köppen ![ORCID],[1] Joan Kleine,[1] Matthew D McHugh ![ORCID],[2] Walter Sermeus ![ORCID],[3] Linda H Aiken[2]

¹Department of Healthcare Management, Technische Universität Berlin, Berlin, Germany
²Center for Health Outcomes and Policy Research, School of Nursing, University of Pennsylvania, Philadelphia, Pennsylvania, USA
³Institute for Healthcare Policy, KU Leuven, Leuven, Vlaams Brabant, Belgium

**Correspondence to**
Dr Claudia B Maier;
c.maier@tu-berlin.de

## ABSTRACT

**Introduction** Many countries in Europe are facing a shortage of nurses and seek effective recruitment and retention strategies. The nursing workforce is increasingly diverse in its educational background, ranging from 3-year vocational training (diploma) to bachelor and master educated nurses. This study analyses recruitment and retention strategies for academically educated nurses (minimum bachelor), including intention to leave, job satisfaction and work engagement compared with diploma nurses in innovative German hospitals; it explores recruitment and retention challenges and opportunities, and identifies lessons on recruitment and retention taking an international perspective.

**Methods and analysis** The study will apply a convergent mixed-methods design, including qualitative and quantitative methods. The qualitative study will include semistructured interviews among hospital managers, nurses, students and stakeholders in Germany. In addition, expert interviews will be conducted internationally in countries with a higher proportion of bachelor/master nurses in hospitals. The quantitative, cross-sectional study will consist of a survey among professional nurses (bachelor/master, diploma nurses) in German hospitals. Study settings are hospitals with a higher-than-average proportion of bachelor nurses or relevant recruitment, work environment or retention strategies in place. Analyses will be conducted in several phases, first in parallel, then combined via triangulation: the parallel analysis technique will analyse the qualitative and quantitative data separately via content analyses (interviews) and descriptive, bivariate and multivariate analyses (survey). Subsequently, data sources will be collectively analysed via a triangulation matrix focusing on developing thematic exploratory clusters at three systemic levels: microlevel, mesolevel and macrolevel. The analyses will be relevant for generating lessons for clinical nursing, management and policy in Germany and internationally.

**Ethics and dissemination** Ethics approval was obtained by the Charité Ethics Committee.
Several dissemination channels will be used, including publications and presentations, for the scientific community, nursing management, clinical nurses and the wider public in Germany and internationally.

---

## STRENGTHS AND LIMITATIONS OF THIS STUDY

⇒ Mixed-methods study in selected German hospitals on recruitment and retention for academically educated nurses (bachelor/master) compared with diploma nurses.
⇒ Qualitative study with in-depth insights, covering managers, nurses, students, stakeholders and international experts.
⇒ Quantitative survey to quantify the intention-to-leave among bachelor/master educated nurses compared with diploma nurses (primary endpoint) and work engagement, job satisfaction (secondary endpoints).
⇒ One strength is the focus on innovative hospitals with a comparatively high proportion of bachelor (and higher) educated nurses and/or relevant recruitment and retention strategies in place, a limitation is that it is not representative of all German hospitals.
⇒ The cross-sectional design is a limitation as to the causality of the findings.

## INTRODUCTION

Several countries in Europe and globally are facing a nursing shortage which has been exacerbated by the COVID-19 pandemic.[1] Effective recruitment and retention strategies for this workforce are therefore of high relevance in hospitals and other healthcare settings in many countries.[2] At the same time, the nursing workforce has become more diverse in its educational background, comprising increasing proportions of nurses with a bachelor of science in nursing (BSN), master of science in nursing (MSN) and in some countries, even nurses with a doctorate degree. This has been a result of nursing educational reforms over the past two decades often triggered by the Bologna process in Europe.[3 4] While some countries have moved the primary nursing education entirely to bachelor level, others have bachelor and

vocational education coexisting, for example, in Germany and several other countries.[4]

A wide range of interventions influence the recruitment and retention of nurses and other health professionals.[2 5 6] A literature review of 23 studies found that a positive, relational leadership style and supportive work environments were associated with nurses' intention to stay in their current position.[5] A review and multiple case study approach[2] on recruitment and retention for health professionals in Europe suggested that bundles of interventions were more effective than single measures but based on few high-quality evaluations across Europe. Successful strategies to attract and retain newly qualified nurses were identified as internship or residency programmes, orientation and transition to practice programmes with a mentor component, while the length varied considerably.[6] Most reviews either subsumed all health professionals[2] or focused on nurses,[5 6] yet there is limited research on recruitment and retention for bachelor and higher educated nurses compared with diploma nurses with a 3-year vocational training.

Intention-to-leave (ITL) is a common indicator to measure nurses' willingness to leave the current employment or the nursing workforce. It can also serve as a proxy for retention in hospitals. At the management level, supportive leadership and work environment have been associated with increased retention.[5] At the individual nurse level, commitment to the job, resilience, levels of knowledge and confidence, as well as preparation for transition and the expectations regarding supervision were identified as relevant. In the USA, reasons to stay with the current employer were autonomy, payment and organisational policies, among others.[7] A theoretical model by Cowden and Cummings identified four key areas associated with intention to stay, namely management (eg, leadership, recognition, shared decision-making), organisation-specific characteristics (eg, career development, staffing), work characteristics (eg, autonomy, work group cohesion) and individual nurse characteristics (eg, education, age).[8]

Internationally, few studies have compared ITL among bachelor (or higher) educated nurses compared with non-academically educated nurses. A survey in the USA found that the higher the educational level (from diploma to doctorate) the lower was the ITL of the nursing profession within the next year.[9] In another US study, ITL was 21% of all nurses surveyed in hospitals,[10] but the probability was slightly higher for BSN or higher qualification. Major differences existed across generations. The probability for ITL was higher for generation Z (born 1997 and later) but lower for Generation X (born 1965–1980) and Boomer (born 1946–1964) in contrast to Millennials (born 1981–1996; reference category) (ibid). Results of a study in South Korea[11] showed that BSN had a higher probability for ITL than nurses with a master's degree or higher. In addition, ITL was higher for younger nurses (26–40 years vs 41 years and older), nurses with less years of work experience and nurses working in smaller hospitals (<400 vs ≥800 beds). A study from China among newly graduated nurses showed that the ITL was 6,7%, with a higher proportion of BSN versus diploma nurses.[12]

To date, there has been limited research in Germany on academically educated nurses (minimum bachelor) in hospitals and their recruitment and retention. One of the reasons is the small number of nurses with at least bachelor degree in direct patient care, which was estimated to be less than 1% in all settings[13] and 2% in 2018 in university hospitals,[14] which is very low compared with many other countries worldwide.

In Germany, one cross-sectional study among 273 nurses with at least bachelor degree found high satisfaction rates but a risk of losing these nurses from direct patient care, as approximately 70% reported being enrolled or planning to enrol in additional studies.[15] A position paper[16] suggested that a specific strategy for bachelor nurses' employment may be a facilitating factor to attract and retain this workforce.

In Germany, there are two major educational pathways to become a professional nurse: either a 3-year vocational training in nursing (diploma) or BSN. The 3-year training has been the predominant educational pathway to become a professional nurse in Germany and is usually offered by nursing schools linked to hospitals. Bachelor education takes 3.5 to 4 years at universities of applied science or universities. The successful completion of either the 3-year training or the bachelor graduation including a state exam are the requirements to work as a professional nurse, also recognised by the European Union.[17] BSN are considered professional nurses and officially have no additional roles compared with those of a professional nurse with a 3-year training, however, some hospitals use individual approaches to employ and integrate BSN in practice. Subsequent studies at Master's level (MSN) can be pursued in the field of clinical practice with various subspecialisations (including in advanced practice nursing, community health nursing), in nursing management, education or nursing science.

Other countries have shown to have more career pathways and options for BSN or MSN in clinical care.[18] In light of the current nursing shortage in Germany,[19] identifying strategies to recruit and retain BSN/MSN is important for German hospitals. Moreover, previous research has shown that higher proportion of BSN/MSN are associated with improved quality of care, lower mortality, improved job satisfaction and intention to stay.[20–22]

The study BSN4Hospital aims to analyse strategies and factors that influence the retention and recruitment of BSN/MSN in comparison with diploma nurses in a selection of hospitals in Germany, including ITL, job satisfaction and work engagement. It also covers the in-depth perspectives of managers, staff, students and stakeholders in Germany. Moreover, the study will have an international component, it will identify international lessons from countries with a longer tradition of BSN/MSN in clinical care. The overall aim is to identify measures

amenable to change to retaining and attracting academically qualified nurses in clinical practice.

## METHODS AND ANALYSIS

BSN4Hospital (funded by the German Federal Ministry of Education and Research, 2021–2024) will apply a convergent mixed-methods study, including a qualitative and quantitative research design. The study follows the conceptual approach by Cowden and Cummings, namely management characteristics, organisation-specific, work and individual nurse characteristics.[8]

Study settings are hospitals with (1) a higher-than-average percentage of nurses with at least BSN education which was estimated to be less than 1% in all hospitals/healthcare settings[13] or hospitals with comparatively high number of BSN as per the 2020 nursing budgets submitted to the German Institute for the remuneration in hospitals[23] and/or (2) having relevant strategies in place to increase or retain nurses with BSN/MSN. Relevant strategies are defined as either single or combined recruitment and retention strategies or broader measures to enhance the work environment. Examples of hospitals to improve the work environment are hospitals participating in the EU-funded Magnet4Europe study.[24] The study focuses on the implementation of the US Magnet model[25] in over 60 hospitals in six European countries, including in Germany.[24] Within Germany, 20 German hospitals participate to improve nurses' work environment, including increasing the percentage of academically educated nurses.

### Patient and public involvement

While there is no direct patient or public involvement planned, the qualitative design of the study will cover patient representation. Most patients are probably not familiar with BSN/MSN due to the relatively low numbers on hospital wards. Yet, patient representatives are expected to be more familiar with topics on recruitment and retention and academisation of nursing, and therefore, will be recruited via the German federal patient organisation.

### Qualitative research

The qualitative design of this study will include different target groups to allow for the various perspectives involved in recruitment/retention: hospital managers to cover both leadership and management perspectives; BSN/MSN to cover those directly involved in patient care; students as the next generation of nurses; and stakeholders in Germany and internationally on contextual and policy factors.

The following target groups will be covered: (1) management perspective: chief nursing officers (CNO), nurse managers and other key resource persons to focus on what managerial strategies are in place, lessons for implementation and uptake in practice and (2) BSN/MSN as well as nursing students' working in patient care

on effective measures, barriers and facilitators in clinical care. Moreover, experts, key stakeholders from nursing associations or organisations and the federal German patient representative organisation will be interviewed on the main facilitating and hindering factors at policy level. Finally, interviews will also be conducted with international experts (eg, hospital CNOs from Belgium, the USA and other countries) to generate lessons internationally and for Germany. The inclusion of international experts in the qualitative design will generate insights into recruitment and retention challenges and lessons in different healthcare systems. The aim is to learn from countries more advanced than Germany with employing BSN and MSN. We chose Belgium and the USA (and may extend to other countries) because these countries have a longer history of BSN and more nursing research (than Germany), but the countries still have the coexistence of BSN and diploma nurses/nurses with vocational training in hospitals which is similar to Germany.

The interviews will follow semistructured interview guides, which have been developed and tailored to the different target groups. The questions in the interview guides were designed as open-ended questions for each of the identified target groups and themes, starting each with an introductory question and several probing questions to obtain additional insights, as necessary. All interview guides are in German, except the interview guide for international experts which was developed in English (see online supplemental file 1). The interview guide for international experts differs in several aspects from the other interview guides. It refers to other countries' experiences, lessons and good practices. Questions will be asked about what strategies and measures have been taken in these countries, with which results and lessons for implementation. The perspectives by international experts can provide valuable comparative insights, which may go beyond what would be generated within Germany alone and may show potential transferable strategies of benefit for German hospitals or policy; as well as for other countries with a coexistence of BSN/MSN and diploma nurses.

In addition to the interview guides, a short survey was developed in German and English to obtain information about the interviewees' background (eg, title/function and job position, years of work experience, educational background, year of birth). Qualitative data collection will follow the Consolidated criteria for reporting qualitative research (COREQ guidelines), which include 32 good practice criteria for interviews and focus groups.[26]

### Sampling strategy and data collection

The interviews will be conducted in at least six German hospitals following the inclusion criteria. The sampling strategy will use a mix of purposeful sampling to select hospitals with contrasting strategies to maximise variation (eg, bundle of measures vs single measures, measures targeted at academically educated nurses and for subgroups of BSN/MSN). The sampling of national and

international experts will also be based on purposeful sampling, combined with network strategies, which is based on the researchers' national and international networks as well as hospital networks. We aim to conduct at least 30–40 interviews, but we will extend the number of interviews until data saturation is achieved.

The minimum number of interviewees by target group is as follows: a minimum of 6 interviews with hospital CNOs or other managers (management perspective), minimum of 12 interviews with BSN/MSN working in direct patient care, including nursing students (nurses' perspective), as well as 10–12 interviews with German as well as international stakeholders and experts.

The interviews will be conducted by researchers of the project team who have been trained in conducting interviews and have prior experience in conducting qualitative research. Interviews will take place in person or digitally, depending on the interviewee's availability. Participants will be informed about the aims and contents of the study, data protection and ethical considerations, and the interview procedure. All participants will be asked to sign a consent form prior to the interview. Recordings will be made with a dictaphone (if in person) or via Zoom (if digital, via end-to-end encryption) provided permission is given. Interviews will be transcribed verbatim by a company with expertise in interview transcriptions. The interviews are expected to last between 45 and 90 min.

### Quantitative study

An overview of the key variables and instruments of the survey are available in table 1.

The following endpoints will be measured:

Primary outcome measure

▶ ITL, this item rates the likelihood 'to leave the current hospital within the next year as a result of job dissatisfaction' ('yes/no' and if 'yes', the follow-up question: 'if yes, what type of work would you seek? (A) nursing in another hospital, (B) nursing, but not in a hospital or (C) non-nursing').

Secondary outcome measures

▶ Job satisfaction, using a single ordinal scale asking, 'How satisfied are you with your current job?' (1 'very dissatisfied' to 4 'very satisfied').

▶ Work engagement score based on the Utrecht Work Engagement Scale-3 using a short version of three indicators measuring vigour, dedication and absorption.

Additional variables to be included:

**Table 1** Topics and instruments of the BSN4Hospital questionnaire

| Topics | No of questions/ items (+ follow up) | Validated instrument (yes/no) | Name of instrument | Source |
|---|---|---|---|---|
| Primary outcome measure | | | | |
| Intention to leave* | 1 (+1) | Yes | RN4CAST | (1) |
| Secondary outcome measures | | | | |
| Job satisfaction | 1 | Yes | RN4CAST | (1) |
| Work engagement | 1/3 | Yes | UWES-3 | (2) |
| Additional variables | | | | |
| Role clarity, job control, performance feedback | 1/9 | Yes | NQPS, QEEW | (3) |
| Leadership | 1/12 | Yes | – | (4) |
| Team involvement | 1/2 | Yes | – | (5) |
| Last shift | 1 | Yes | RN4CAST | (6) |
| Quality and safety | 3 | Yes | RN4CAST | (6) |
| Recruitment: Reasons for choosing hospital | 1/11 | No | | (7) |
| Retention: Reasons to stay (minimum 5 years) | 1/19 | No | | (7) |
| Career opportunities in hospital | 1/7 | No | | (7) |
| Expanded roles and tasks | 1/9 | No | | (8) |
| Work attitudes by generation (baby boomer, X, Y, Z) | 1/4 | No | | (9) |
| Sample characteristics (eg, age, gender, contractual situation) | 12 (+1) | No | | |
| Educational background | 7 (+3) | No | | |

*Wording of the question: 'If possible, would you leave your current hospital within the next year as a result of job dissatisfaction? If yes, what type of work would you seek?'. Source: (1) RN4CAST,[35] (2) UWES-3,[36] (3) NQPS, QEEW,[37 38] (4) Schaufeli and Rahmadani et al,[39 40] (5) Schaufeli,[39] (6) Sermeus et al,[35] (7) developed and modified from,[2 15 41] (8) developed and modified from,[42 43] (9) developed and modified from.[44 45]

NQPS, Nordic Questionnaire for Psychosocial Factors at work; QEEW, Questionnaire on the Experience and Evaluation of Work; RN4CAST, Registered Nurse Forcasting; UWES-3, Utrecht Work Engagement Scale.

Role clarity and job control, leadership, team involvement, staffing (self-reported), quality and safety (self-reported), recruitment: reasons to choose a hospital as employer ('ideal hospital') (eg, reputation, good team work, good work-life balance, high job security, pay by educational qualification, possibility to opt-out from night-shifts, among others), retention: reasons to stay more than 5 years in a hospital (eg, work-life balance, team work, nurse staffing, autonomy in daily work, career possibilities, flexible work hours or possibility for sabbatical, flat hierarchy, supportive leadership), career opportunities, work-life priorities by generations (Boomer, X, Y, Z), sociodemographic information (age, sex), average length of employment in hospital (in years) and information about the educational background of respondents (BSN, MSN, doctorate, additional non-academic specialisations).

The survey was piloted internally and externally in 2022: internally three colleagues with expertise in conducting surveys and with a background in nursing or physiotherapy answered the survey and provided in-depth feedback and comments. Externally, three nurses working in one hospital in Berlin to which personal contacts existed, provided feedback and comments. Moreover, members of the advisory board of the project provided additional feedback on the survey. The feedback was integrated in subsequent rounds.

## Sample size

The main endpoint in the quantitative part of the BSN4Hospital study is the ITL (vs intention to stay) the current hospital. To date, no study in Germany has analysed ITL between BSN/MSN and diploma nurses using statistical methods, and therefore, no effect sizes exist for Germany. In the international literature, two studies[9] [11] described small effect sizes for ITL in relation to the qualification level, Cohen's d 0.15[11] and 0.1.[9] For the calculation of the sample size, therefore, we chose an effect size of 0.2 (Cohen's d), which indicates small differences between two groups and is considered relevant. The rationale for this is that recruitment and retention measures should be adapted even for small differences in the ITL between nurses with and without academic degrees. In addition, a test power of 80% was chosen, a significance level of 5%, and two-sided hypothesis. The sample size calculation took into account the small number and percentage of BSN/MSN in German hospitals and was estimated at 2% for university hospitals[14] and 1% for all other hospitals.[13]

As the basis for the sample size calculation, the data on full-time staff were retrieved from the 2019 quality reports of hospitals[27] for the 20 hospitals participating in Magnet4Europe. The proportion of nursing professionals working part time was 49% in 2020.[28] On this basis, the number of full-time staff was calculated to number of persons. Following the 2.1% value for university hospitals,[14] the mean value for the nine participating university hospitals is 42.3 BSN/MSN (42.3×9 university

hospitals=380.7 nurses) and in general hospitals 6.6 BSN/MSN (6.6×11 hospitals=72.6 nurses; estimated 1% value of BSN/MSN). This leads to a total of 453.BSN/MSN in 20 hospitals.

A response rate of 40% to 60% is assumed. With an estimated number of 453 BSN/MSN in the 20 hospitals, between 181 individuals (40% response rate) and 272 individuals (60% response rate) are expected to participate. Assuming an effect size of 0.2 (with a test power of 80%, a significance level of 5% and two-sided hypothesis generation), a sample of 250 individuals with academic degrees will be needed. If the response rate is only 40%, additional hospitals will be invited to join the study until at least 250 participants with academic degree (minimum bachelor level) are reached. For the second group (diploma nurses), a sample size of 940 is calculated. If a response rate of 40%–60% is also assumed for the diploma nurses, between 2350 (40% response rate) and 1567 (60% response rate) would have to be invited to the survey. These values are considered achievable since a total of 21 700 nursing professionals (full-time staff) are employed in the participating hospitals according to the quality reports. The ratio of 250 : 940 (2.5/9.4) results from the fact that a very small proportion of nurses in Germany have an academic degree. Although a response rate of 40%–60% is considered high, it is assumed that the study is feasible and that the response rate will be achieved. This is based on the following assumptions: first, the 20 hospitals participating in Magnet4Europe are motivated to improve nursing and their work environments in general and focus on recruitment and retention for all nurses and BSN/MSN. Second, some hospitals are specifically working towards increasing the proportion of BSN/MSN. Third, there is already a well-established contact between the hospitals and the researchers at TU Berlin. Finally, it is assumed that BSN/MSN are motivated to participate in the online survey, especially since it deals with topics such as ITL vs intention to stay, work engagement, job satisfaction, among others.

## Data collection

The survey will be conducted online (1) in hospitals participating in the Magnet4Europe study and (2) in additional hospitals with a comparatively higher proportion of BSN/MSN than the average estimated 1%–2%. Data collection is planned for the second and third quarter of 2023. Hospitals will be asked to identify all wards with at least one BSN/MSN, subsequently, all professional nurses with a minimum of 3-year qualification (vocational education, BSN, MSN, etc) are eligible to participate. Excluded are nursing assistants and non-nursing professionals. Excluded are also psychiatry/mental health wards, because psychiatric care differs substantially from somatic care and was therefore not within the scope of the study.

The survey will be conducted via SoSciSurvey,[29] an online survey tool for scientific studies. The objectives of the study will be communicated by the project coordinators of the individual hospitals to the nurses (via posters

or flyers), this also includes the URL and further information on how to participate in the survey. Interested nurses can thus participate in the survey directly via the URL. Informed consent to conduct the survey will be obtained from participants prior to the survey; participation without consent is not possible. Each hospital will receive an individual survey link. The survey will be anonymous. In SoSciSurvey, IP addresses are not recorded by default and the option to send reminders is not possible.

### Analysis and triangulation of data

The convergent design is chosen as the theoretical framework construct for this mixed-methods study, which aims to collect two different, yet equally relevant data sources that are complementary.[30][31] Thus, it aims to ensure that the strengths of the quantitative and qualitative data are being used in a complementary, and mutually beneficial way.[31] In the convergent design, the data are often collected simultaneously or shortly after each other. This will also be done for the BSN4Hospital study.

The analysis and triangulation of the data will follow the parallel analysis approach in mixed-methods designs, which first analyses the data separately, and then subsequently via triangulation.[30]

### Qualitative study: analysis of interviews

The interviews will be analysed following Mayring's qualitative content analysis[32] using a software for qualitative data analysis (eg, Atlas.ti) in an iterative way. The material will be analysed using the combined, deductive-inductive principle of categorisations and code development. The deductive phase will apply a set of predefined major codes to the material (eg, recruitment, retention, hospital-level strategies, individual factors, policy level), whereas as part of the inductive phase, new categories will be developed in an iterative process based on the interview material. At least two coders will develop the coding rules and major codes (including inclusion and exclusion criteria) on which basis the major codes and subcodes will be identified. The process will be documented with a codebook and notes for each interviewee.

### Quantitative study: analysis

Statistical analyses will be performed using a statistical software package (for example, Stata version 18 or SPSS version 27), involving descriptive, bivariate and multivariate regression to analyse nurses' ITL (primary endpoint) by qualification level (academic vs non-academic, plus subgroup analyses) and the additional outcome measures job satisfaction and work engagement (secondary endpoints), adjusting for potential confounders (hospital type, bed size, work environment, etc). The analyses will follow the Strengthening the reporting of observational studies in epidemiology statement (STROBE guidelines) for reporting of cross-sectional, observational studies.[33]

### Triangulation of the data

The triangulation of the two data sources will follow the 'parallel analysis' method, that is, first conducted separately for the quantitative and qualitative data.[30] This method of analysis is the most common method described in the literature within mixed-methods studies. In a subsequent step, the data are then combined and compared with provide a comprehensive overview of the findings from different perspectives (qualitative: the in-depth insights and experiences of individual nurses, as well as managers and experts and quantitative: large number of academically educated nurses in selected hospitals). The process of triangulation will involve an iterative approach, first, comparing the quantitative and qualitative data with each other for similarly vs contrasting findings and reinforcing or complementary findings. For this purpose, a triangulation matrix will be created.[34] The aim is to contextualise the results of the quantitative analysis with findings from the qualitative study; and to understand the qualitative data in the context of the questionnaire survey, to explore in depth issues of implementation and recommendations for practice. This will be done according to thematic exploratory clusters at three systemic levels: micro, meso and macro level, defined as individual, management, policy levels. This method of analysis will be particularly relevant for generating recommendations for management and policy-makers.

### Ethics and dissemination

Ethics approval was obtained from the Charité Ethics Committee (EA2/097/22). Participation is anonymous and voluntary and comes with no benefits for the respondents or interviewees. Neither the hospital nor the researchers will know who took part. The interviews will be pseudonymised so that no conclusions about individuals can be drawn.

The following dissemination activities are planned: The findings from the interviews and the survey will be published in scientific and non-scientific journals aimed at various target groups, including the scientific community, policy-makers, nursing managers and the nursing profession.

In addition, the results of the study will be presented at national and international conferences. International conferences with an interdisciplinary focus will be targeted as well those with a focus on nursing. Dissemination will also involve the management levels of hospitals (CNOs, nursing ward managers, board of directors). To this end, recommendations for nursing management and practice will be developed. These will be made available to participating hospitals and to nursing management associations with a request to disseminate within their networks. In order to address a broader non-scientific public in Germany, short articles or press releases will be written and distributed.

In conclusion, this study is expected to be one of the largest studies on academically qualified nurses in German hospitals on recruitment and retention using a convergent mixed-methods design. The study seeks to inform hospital management, clinical nurses and policy-makers on strategies to increase and sustain this workforce. In

addition, it will identify international lessons and policy implications on recruitment and retention of nurses by educational qualification in hospitals.

**Contributors** CBM planned the study and methods and wrote the first draft and subsequent versions of the manuscript. JKö and JKl contributed to the planning of the study, wrote and revised sections of the manuscript. MDM, WS and LHA provided guidance for the study, revised sections and provided overall advice for the manuscript. All authors were responsible for approving the final manuscript.

**Funding** This research received funding by the German Federal Ministry of Education and Research, grant number 01GY2004. Open access funding was provided by Technische Universität Berlin.

**Competing interests** None declared.

**Patient and public involvement** Patients and/or the public were not involved in the design, or conduct, or reporting, or dissemination plans of this research.

**Patient consent for publication** Not applicable.

**Ethics approval** Ethics approval was obtained from the Charité Ethics Committee (EA2/097/22).

**Provenance and peer review** Not commissioned; externally peer reviewed.

**ORCID iDs**
Claudia B Maier http://orcid.org/0000-0002-7734-2258
Julia Köppen http://orcid.org/0000-0001-7941-641X
Matthew D McHugh http://orcid.org/0000-0002-1263-0697
Walter Sermeus http://orcid.org/0000-0002-5915-1845

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
