## [Reviewer comments · BMJ Open]

ARTICLE DETAILS

TITLE (PROVISIONAL)	Recruiting and retaining Bachelor qualified nurses in German hospitals (BSN4Hospital): Protocol of a mixed-methods design
AUTHORS	Maier, Claudia B.; Köppen, Julia; Kleine, Joan; McHugh, Matthew; Sermeus, Walter; Aiken, Linda

VERSION 1 – REVIEW

REVIEWER	Seah, Betsy National University of Singapore
REVIEW RETURNED	17-Apr-2023

GENERAL COMMENTS	Thank you for the opportunity to review this interesting paper that addresses the recruitment and retention of Bachelor qualified nurses. While this paper seeks to provide a comprehensive understanding towards this important topic, it requires further clarification of methods, organisation of content and an elaboration on significance of this study in relation to international readership. Below are the specific comments: Major: a) Abstract: Please reorganise and state the aims of this study explicitly. Aim of quantitative study was mentioned but not the qualitative study. Perhaps it will be good to mentioned that this is a convergent mixed-methods study. b) Abstract: May I suggest mentioning about the “expert interviews internationally” in a separate sentence and provide further elaboration. It can be confusing for the reader in understanding why international experts are interviewed. c) Abstract: May I suggest using the term ‘study settings’ instead of “inclusion criteria” when described about the hospitals with a higher than average proportion of Bachelor nurses... d) It is confusing what is the overall aims of the proposed study and specific objectives of the qualitative and quantitative study, especially when they are only mentioned at different parts of the paper. The reader has to refer back and forth. Please explicitly state them together, the overall aims followed by the specific objectives in the last paragraph of the introduction, prior to the methods section. This will give readers an overview and direction of this study and appreciate the coherence of methods for the entire study. Please also ensure consistency of the aims in the abstract and the main text. e) As not all countries have diploma nurses, it will be helpful if the authors could provide further contextual information on the differences in the training program, as well as the roles and responsibilities of a Bachelor qualified nurse verses a diploma qualified nurse. Also, it might sound discriminatory when diploma graduated nurses are termed as ‘no academic education diploma’.
---

	f) Is there an overarching theoretical or conceptual framework that guides the approach of this mixed-methods study? How does the team decide on which identified themes or variables to explore and examine? g) Why did the team decide to include the different target groups for the qualitative study? h) It is unclear what is the intent of including international experts for the qualitative study, especially if the aim of this study was to 'analyse hospital-level strategies and factors that influence the retention and recruitment of academically qualified nurses (minimum Bachelor) in comparison with nurses with no academic education (diploma) in a selection of 'innovator' hospitals in Germany.' Also, how does the interview guide for international experts differ from that for the rest of the topic groups, especially in relation to the identified themes. i) Noted incomplete sentence 'patient and public involvement'. This is a new concept introduced in the methods. Kindly provide some contextual information regarding the significance of including patient representatives and how they are approached and recruited. j) It was mentioned that at least 30 interviews would be conducted. Does this refer to each target group, or for all target groups? How many participants will be recruited from each target group? k) It is unclear who will be conducting the qualitative interviews and the relationship with the participants. l) Why are nurses working in psychiatry or mental health wards excluded for the quantitative study? m) Kindly elaborate what are some actions taken to ensure the ethical n) It will be good if the authors could highlight and strengthen the significance of this study, and its specific potential contributions to the literature, nursing management, practice and policy at an international level, rather than pitching at Germany as a country. Minor a) It is confusing when the inclusion criteria are mentioned twice, once for the participating hospitals, and the other for the individual participants. May I suggest removing the words 'inclusion criteria' in the second paragraph of the methods section. b) Content in the methods section seems choppy and some of the information provided is repetitive. Please review the flow of the paper and avoid repetition of information e.g.,  • "in addition, other hospitals with high percentages of academically qualified nurses also qualify to be included" • mentioned twice about COREC guidelines
--	---

REVIEWER	Schmollgruber, Shelley University of the Witwatersrand Faculty of Health Sciences, Nursing Education
REVIEW RETURNED	19-Apr-2023

GENERAL COMMENTS	Thank you for the opportunity to read your study proposal. I did find the proposal well written and interesting to read. This is a good piece of scholarly work. I have found a few areas that need some explanation or clarification. A separate document is attached for the authors' attention. Reviewer Comments: The authors want to investigate the retention, job satisfaction and work engagement between academic nurses (Bachelor/Master) and diploma-educated nurses. The emphasis is to compare
--

specific data sources between the two groups. Its purpose is to quantify outcomes and influencing factors amongst these nurses in a selection of "innovator" hospitals in Germany. This is a well-written protocol and an important topic related to workforce planning for hospitals, nurses and researchers to explore.

Abstract

The abstract is well-written and follows the scientific format. However, on page 1, lines 22 to 28), the aim of the study appears after the methods and analysis have been described. My thinking is that the aim of the study is achieved through the methods and analysis and not the other way around. Maybe the authors could clarify or reconsider this thinking.

Introduction

This is a well-written section and flows well. The background and key concepts informing the study are well described. The authors have explained from a global perspective the shift of nursing from diploma to degree-based education and applied key issues to the local context where they intend to conduct the study.

Terminologies

Although this is a well-written proposal, my concerns are the variations in terminologies when referring to "degreed nurses as academically educated" and 'diploma nurses as non-academically trained (page 6, line 50) or vocational education (page 4, line 18-19). I am concerned that authors refer to diploma nurses as non-academically trained or vocational. Some readers might be discouraged from reading this study, as diploma-level training is the only option for nurses in their countries. This could detract from reading this important and carefully structured work. I would suggest the authors consider using – degree (BSN, MSN) and non-degree (diploma) terms when describing these groups.

On page 8 (lines 40 to 45) it is also stated that "BSN, MSN and APN will be recruited. ??? APNs are these diploma/non-academically trained nurses / or academically trained nurses ??? Please check the entire manuscript to ensure clarity/ consistency when referring to these terms.

Methods and analysis

Even though this section is clear and well-written some clarification is needed. For example, In target group 4, can the authors clarify who these key stakeholders are in this study (page 8, lines 6-7)? Also, can the authors clarify what information they seek to elicit for target groups 4 and 5 as they did for target groups 1 to 3)? Readers might be interested in this aspect of the proposal.

Methods related to the study's qualitative and quantitative aspects are correct and well applied.

Analysis

This will be conducted in several phases, first parallel, then combined via triangulation. The parallel analysis technique will separately analyse the qualitative and quantitative data via content analysis (interviews) and descriptive, bivariate and multivariate

	analysis (survey). Subsequently, data sources will be collectively analysed via a triangulation matrix focusing on developing thematic exploratory clusters at three systematic levels (micro, meso and macro level). The interviews will be analysed following Mayring's content analysis using software for qualitative data analysis (Atlas. ti). This analysis will be relevant for generating lessons on clinical nursing, management and policy. Data collection Data collection is expected to comment in the latter part of 2023 using an online survey. Inclusion criteria are well defined. Exclusion criteria are not justified (page 12). Can the authors comment briefly on why nursing assistants, non-nursing professionals and psychiatry/mental health wards have been excluded (page 12, lines 58 to 60)? Conclusion The study is well-motivated and justified by the authors. In their opinion, this is the first study and is expected to be one of the most extensive studies in Germany on this topic. The outcomes will interest other researchers worldwide who may face similar challenges in transforming the nursing workforce. The proposal demonstrates the scholarly aspects expected within a mixed methods study. I think that the study is well-constructed and highly feasible.
--	---

VERSION 1 – AUTHOR RESPONSE

Reviewer: 1

Dr. Betsy Seah, National University of Singapore

Comments to the Author:

Thank you for the opportunity to review this interesting paper that addresses the recruitment and retention of Bachelor qualified nurses. While this paper seeks to provide a comprehensive understanding towards this important topic, it requires further clarification of methods, organisation of content and an elaboration on significance of this study in relation to international readership.

Authors' response: Thank you for taking the time to review our paper and for the constructive feedback. The comments have been very helpful and we believe the paper has considerably improved.

Below are the specific comments:

Major:

- a) Abstract: Please reorganise and state the aims of this study explicitly. Aim of quantitative study was mentioned but not the qualitative study. Perhaps it will be good to mention that this is a convergent mixed-methods study.

Authors' response: Thank you for this observation. We have revised the aims of the study to also take into account the aims of the qualitative research. Also, we refer now in the abstract to the convergent mixed-methods design.

b) Abstract: May I suggest mentioning about the "expert interviews internationally" in a separate sentence and provide further elaboration. It can be confusing for the reader in understanding why international experts are interviewed.

Authors' response: Thank you, yes, we followed your advice.

c) Abstract: May I suggest using the term 'study settings' instead of "inclusion criteria" when described about the hospitals with a higher than average proportion of Bachelor nurses...

Authors' response: Thank you for the suggestion, which we followed.

d) It is confusing what is the overall aims of the proposed study and specific objectives of the qualitative and quantitative study, especially when they are only mentioned at different parts of the paper. The reader has to refer back and forth. Please explicitly state them together, the overall aims followed by the specific objectives in the last paragraph of the introduction, prior to the methods section. This will give readers an overview and direction of this study and appreciate the coherence of methods for the entire study. Please also ensure consistency of the aims in the abstract and the main text.

Authors' response: We have followed your advice, thank you.

e) As not all countries have diploma nurses, it will be helpful if the authors could provide further contextual information on the differences in the training program, as well as the roles and responsibilities of a Bachelor qualified nurse versus a diploma qualified nurse. Also, it might sound discriminatory when diploma graduated nurses are termed as 'no academic education diploma'.

Authors' response: Thank you for this helpful comment. We have removed references to nurses with "no academic education" and use the term "diploma nurses" instead. We have also added a section under Background and describe the educational systems for nurses in Germany: entry into nursing is possible via two pathways: either the 3-year vocational training ("diploma nurses") or through Bachelor programmes at universities of applied sciences (or universities). Both lead to the title of a professional nurse, also recognized under the EU directives. There are no other officially defined roles for BSN in Germany other than those of a professional nurse, but some hospitals have defined their own roles for BSN going beyond those of a nurse with a 3-year training.

f) Is there an overarching theoretical or conceptual framework that guides the approach of this mixed-methods study? How does the team decide on which identified themes or variables to explore and examine?

Authors' response: Thank you. We follow the conceptual approach by Cowden & Cummings who identified four areas relevant to intention-to-stay vs. intention-to-leave: management, organizational characteristics, work characteristics, individual nurse characteristics. We have included and expanded upon it in the Background and make it more explicit (last sentence in the Background) that we cover these elements.

g) Why did the team decide to include the different target groups for the qualitative study?

Authors' response: Thank you. We have decided to include different target groups in the qualitative study to allow for the various perspectives and levels involved in recruitment/retention: managers and staff to cover both management/leadership perspectives; those directly involved in patient care (BSN/MSN as well as students) and stakeholders on contextual and policy factors (in Germany and internationally). This approach allows us to explore diverse viewpoints and experiences. We have added a short section in the Methods to clarify this approach.

h) It is unclear what is the intent of including international experts for the qualitative study, especially if the aim of this study was to 'analyse hospital-level strategies and factors that influence the retention and recruitment of academically qualified nurses (minimum Bachelor) in comparison with nurses with no academic education (diploma) in a selection of 'innovator' hospitals in Germany.' Also, how does the interview guide for international experts differ from that for the rest of the topic groups, especially in related to the identified themes.

Authors' response: Thank you for this observation. We have revised the section. In fact, the inclusion of international experts in the qualitative study enables us to obtain insights into recruitment and retention challenges and lessons in different country contexts and healthcare systems. The aim is to learn from countries more advanced than Germany with employing BSN and MSN. We chose Belgium and the USA (but may extent to other country experts) and use country selection criteria as having a longer history of BSN (than Germany) and the co-existence of BSN and diploma nurses/those with vocational training in a country.

The interview guide for international experts differs in several aspects from the other interview guides, as it refers to other countries' experiences, lessons and best practices in recruiting and retaining academically qualified nurses in hospitals. Questions will be asked about what strategies and measures have been taken in these countries, with which results and lessons for implementation. The perspectives by international experts can provide valuable comparative insights which may go beyond what would be generated within Germany alone and may show potential transferable strategies of benefit for German hospitals or policy.

i) Noted incomplete sentence 'patient and public involvement'. This is a new concept introduced in the methods. Kindly provide some contextual information regarding the significance of including patient representatives and how they are approached and recruited.

Authors' response: Thank you for this comment. 'Patient and public involvement' is recommended by BMJ Open as a subheading. The approach and necessity to involve patient representatives were added.

j) It was mentioned that at least 30 interviews would be conducted. Does this refer to each target group, or for all target groups? How many participants will be recruited from each target group?

Authors' response: Thank you. This information refers to all target groups and is the minimum number, we may well be above the minimum and have therefore included a range of at least 30-40 interviews, which is as a breakdown: min. 6 managers/CNOs of the hospitals, min. 12 interviews with BSN/MSN/students working in clinical care, as well as a minimum of 10-12 interviews with German stakeholders as well as international stakeholders.

k) It is unclear who will be conducting the qualitative interviews and the relationship with the participants.

Authors' response: The interviews will be conducted by researchers within the project team. One has a PhD, two hold a master's degree and are doctoral students with experience in qualitative research, and two are in their final stage of their Master's studies. All have experience in conducting qualitative research. Moreover, all have been trained in conducting qualitative research and have performed previous interviews and a pilot interview for this research. Three researchers know some of the potential interviewees (e.g. CNOs in selected hospitals) through a prior research project (Magnet4Europe), which will be reflected prior, during and after the interviews and described in field notes after the interviews, as necessary.

l) Why are nurses working in psychiatry or mental health wards excluded for the quantitative study?

Authors' response: Thank you. Nurses working in psychiatry perform a set of tasks which can be quite different compared with nurses in somatic care. The survey also covers roles and tasks performed in practice by BSN / MSN and ITL, job satisfaction and work engagement may be associated with that (among others). Due to the length of the survey, it was not possible to cover relevant aspects of psychiatric care. We included a sentence to make clear why we excluded psychiatric wards.

m) Kindly elaborate what are some actions taken to ensure the ethical

Authors' response: Thank you. The remainder part of the sentence was missing, hence, we guess that the request was to elaborate on the actions taken to ensure the ethical conduct of the study. We added the major activities that have been undertaken already/ are planned.

n) It will be good if the authors could highlight and strengthen the significance of this study, and its specific potential contributions to the literature, nursing management, practice and policy at an international level, rather than pitching at Germany as a country.

Authors' response: We have expanded on this important point in the background, the methods and the last paragraph of the protocol.

Minor

a) It is confusing when the inclusion criteria are mentioned twice, once for the participating

hospitals, and the other for the individual participants. May I suggest removing the words 'inclusion criteria' in the second paragraph of methods section.

Authors' response: Thank you. We have followed your advice and refer to study settings for hospital selection, and to inclusion criteria for the inclusion criteria for the survey participants.

b) Content in the methods section seem choppy and some of the information provided are repetitive. Please review the flow of paper and avoid repetition of information e.g.,

- "in addition, other hospitals with high percentages or academically qualified nurses also qualify to in be included"
- mentioned twice about COREC guidelines

Authors' response: Thank you for the observation, we followed the suggestions.

Reviewer: 2

Dr. Shelley Schmolgruber, University of the Witwatersrand Faculty of Health Sciences

Comments to the Author:

Thank you for the opportunity to read your study proposal. I did find the proposal well written and interesting to read. This is a good piece of scholarly work. I have found a few areas that need some explanation or clarification. A separate document is attached for the authors attention.

Authors' response: Thank you for your assessment of the paper and the valuable and constructive comments. We believe that they have considerably helped to improve the paper and its reporting and we are most grateful for your feed-back. Please find our responses provided below.

Reviewer Comments:

The authors want to investigate the retention, job satisfaction and work engagement between academic

nurses (Bachelor/Master) and diploma-educated nurses. The emphasis is to compare specific data sources

between the two groups. Its purpose is to quantify outcomes and influencing factors amongst these nurses

in a selection of "innovator" hospitals in Germany. This is a well-written protocol and an important topic

related to workforce planning for hospitals, nurses and researchers to explore.

Authors' response: Thank you for this overall positive assessment.

Abstract

The abstract is well-written and follows the scientific format. However, on page 1, lines 22 to 28), the aim of

the study appears after the methods and analysis have been described. My thinking is that the aim of the

study is achieved through the methods and analysis and not the other way around. Maybe the authors could clarify or reconsider this thinking.

Authors' response: Thank you. Reviewer 1 also made this comment and we have now moved the aim of the study at the end of the background section.

Introduction

This is a well-written section and flows well. The background and key concepts informing the study are well

described. The authors have explained from a global perspective the shift of nursing from diploma to degree-based education and applied key issues to the local context where they intend to conduct the study.

Terminologies

Although this is a well-written proposal, my concerns are the variations in terminologies when referring to

"degreed nurses as academically educated" and '**diploma nurses as non-academically trained** (page 6, line 50) or **vocational education** (page 4, line 18-19). I am concerned that authors refer to diploma nurses as non-academically trained or vocational. Some readers might be discouraged from reading this study, as diploma-level training is the only option for nurses in their countries. This could detract from reading this important and carefully structured work. I would suggest the authors consider using – **degree** (BSN, MSN) and **non-degree** (diploma) terms when describing these groups.

On page 8 (lines 40 to 45) it is also stated that "**BSN, MSN and APN will be recruited**. ??? APNs are these

diploma/non-academically trained nurses / or academically trained nurses ??? Please check the entire

manuscript to ensure clarity/ consistency when referring to these terms.

Authors' response: Thank you for this observation, which was also alluded to by reviewer 1. We have now added a paragraph on nursing education in Germany, where we refer to BSN / MSN education and the diploma nurses, who have a 3-year vocational training. We now refer to BSN, MSN and diploma nurses wherever possible and have tried to minimize other terms. "Degree nurses" is very unknown to many German readers (even those who speak German), hence we want to stick to BSN and MSN instead of "degree nurses".

APN are advanced practice nurses with usually a minimum Master's degree. We have deleted the term because they are also covered under MSN.

Methods and analysis

Even though this section is clear and well-written some clarification is needed. For example, In target group

4, can the authors clarify who these **key stakeholders** are in this study (page 8, lines 6-7)? Also, can the

authors clarify what information they seek to elicit for target groups 4 and 5 as they did for target groups 1

to 3)? Readers might be interested in this aspect of the proposal.

Methods related to the study's qualitative and quantitative aspects are correct and well applied

Authors' response: Thank you for this advice. We added examples for key stakeholders and provided information why we aim to interview the different target groups.

Analysis

This will be conducted in several phases, first parallel, then combined via triangulation. The parallel analysis

technique will separately analyse the qualitative and quantitative data via content analysis (interviews) and

descriptive, bivariate and multivariate analysis (survey). Subsequently, data sources will be collectively

analysed via a triangulation matrix focusing on developing thematic exploratory clusters at three systematic

levels (micro, meso and macro level). The interviews will be analysed following Mayring's content analysis

using software for qualitative data analysis (Atlas. ti). This analysis will be relevant for generating lessons on

clinical nursing, management and policy.

Data collection

Data collection is expected to comment in the latter part of 2023 using an online survey. Inclusion criteria are

well defined. Exclusion criteria are not justified (page 12). Can the authors comment briefly on why nursing

assistants, non-nursing professionals and psychiatry/mental health wards have been excluded (page 12, lines

58 to 60)?

Authors' response: Thank you. The study would also be interesting for nursing assistants or for other professions, however, the focus is on nurses with Bachelor's/Master's degree; and in comparison with diploma nurses. An extension to other professional groups is beyond the scope of the study (e.g. reaching the sample size). Regarding the comment on nurses working in psychiatry, this observation was also made by reviewer 1 and we have included a sentence to make clear why we excluded psychiatric wards

Conclusion

The study is well-motivated and justified by the authors. In their opinion, this is the first study and is expected to be one of the most extensive studies in Germany on this topic. The outcomes will interest other

researchers worldwide who may face similar challenges in transforming the nursing workforce. The proposal demonstrates the scholarly aspects expected within a mixed methods study.

I think that the study is well-constructed and highly feasible.

Good luck with your manuscript

Authors' response: Thank you again, for your constructive and overall positive feed-back. We appreciate the time you spent in reviewing this protocol.

VERSION 2 – REVIEW

REVIEWER	Schmollgruber, Shelley University of the Witwatersrand Faculty of Health Sciences, Nursing Education
REVIEW RETURNED	14-Jun-2023
GENERAL COMMENTS	Thank you for the opportunity to read the revised version of this paper again. I am sufficiently satisfied that ALL the reviewers comments have been adequately addressed. This is a well written and scholarly piece of work. Good luck with your manuscript.